# Improved Light Harvesting of Fiber-Shaped Dye-Sensitized Solar Cells by Using a Bacteriophage Doping Method

**DOI:** 10.3390/nano11123421

**Published:** 2021-12-17

**Authors:** Sung-Jun Koo, Jae Ho Kim, Yong-Ki Kim, Myunghun Shin, Jin Woo Choi, Jin-Woo Oh, Hyung Woo Lee, Myungkwan Song

**Affiliations:** 1Department of Energy and Electronic Materials, Korea Institute of Materials Science (KIMS), Changwon 51508, Korea; sungun666@kims.re.kr (S.-J.K.); jho83@kims.re.kr (J.H.K.); jinwoo.choi@kims.re.kr (J.W.C.); 2Department of Nano Fusion Technology, Pusan National University, Busan 46241, Korea; 3School of Electronics and Information Engineering, Korea Aerospace University, Goyang 10540, Korea; ygk3373@kau.kr (Y.-K.K.); mhshin@kau.ac.kr (M.S.); 4Bio-IT Fusion Technology Research Institute, Pusan National University, Busan 46241, Korea; 5Department of Nanoenergy Engineering and Research Center of Energy Convergence Technology, Pusan Natuional University, Busan 46241, Korea

**Keywords:** fiber-shaped solar cells, dye-sensitized solar cells, M13 bacteriophage, plasmon resonance

## Abstract

Fiber-shaped solar cells (FSCs) with flexibility, wearability, and wearability have emerged as a topic of intensive interest and development in recent years. Although the development of this material is still in its early stages, bacteriophage-metallic nanostructures, which exhibit prominent localized surface plasmon resonance (LSPR) properties, are one such material that has been utilized to further improve the power conversion efficiency (PCE) of solar cells. This study confirmed that fiber-shaped dye-sensitized solar cells (FDSSCs) enhanced by silver nanoparticles-embedded M13 bacteriophage (Ag@M13) can be developed as solar cell devices with better PCE than the solar cells without them. The PCE of FDSSCs was improved by adding the Ag@M13 into an iodine species (I^−^/I_3_^−^) based electrolyte, which is used for redox couple reactions. The optimized Ag@M13 enhanced FDSSC showed a PCE of up to 5.80%, which was improved by 16.7% compared to that of the reference device with 4.97%.

## 1. Introduction

Harvesting solar energy has been regarded as one of the easiest solutions to collect the promising renewable energy resources for sustainable development. Among various flexible energy harvesting technologies, fiber-shaped dye-sensitized solar cells (FDSSCs), which represent a promising future energy source with flexible and wearable properties, are drawing the attention of many researchers because it is lightweight, inexpensive, simple to manufacture, and flexible [1,2,3,4,5,6,7]. The FDSSCs typically consist of three parts: a photoanode (PA), a counter electrode (CE), and a redox electrolyte. Under illumination, dyes act as photosensitizer, in which photons excite electrons at the highest occupied molecular orbital (HOMO) to the lowest unoccupied molecular orbital (LUMO). The excited electrons are injected into the conduction band of the PA and then diffused through the PA into the electrode to reach the CE through an external circuit. On one hand, iodide (I^−^), a reductive species in the electrolyte, supplies electrons to the oxidized dye and becomes triiodide (I_3_^−^), which in turn gains electrons from the CE to complete the redox couple [8,9].

The electrolyte is one of the most important components of dye-sensitized solar cells (DSSCs) and has a significant impact on the high power conversion efficiency (PCE) in solar cells. The electrolyte in solar cell devices should have the following characteristics: (I) It should be able to regenerate the oxidized dye efficiently. (II) Should not corrode with DSSC components. (III) Should rapidly diffuse charge carriers, improve conductivity, and enable effective contact between the working and counter electrodes. (IV) Absorption spectra of an electrolyte should not overlap with the absorption spectra of a dye [10,11]. Therefore, to satisfy the above functions, bacteriophage-metallic nanostructures, which exhibit prominent localized surface plasmon resonance (LSPR) properties, were integrated with the iodine species (I^−^/I_3_^−^) based electrolyte that has been proven as a highly efficient electrolyte. The metallic nanoparticles (NPs) mainly increase the light-surface plasmon coupling, realizing a plasmonic enhancement effect in photoelectric devices [12]. Hence, incorporating bacteriophage-metallic nanostructures into on unit of solar cells is a promising alternative strategy that can improve the performance of photoelectric devices by inducing the LSPR properties, which is receiving great attention for light scattering, fluorescence, charge transfer, and local field enhancement [13,14,15].

Recently, among the bacteriophage-metallic nanostructures, an M13 bacteriophage has attracted particular attention due to its non-toxic, self-assembly, and specific binding properties. This bacterial virus, M13 bacteriophage, is 880 nm in length and 6.6 nm in diameter, consisting of a single-stranded DNA molecule enclosed by 2700 identical copies of the major coat protein pVIII and capped with five copies of four different minor coat proteins at the ends [16], as shown in Figure 1a. The M13 bacteriophage is easily applicable to wearable applications because it is stable in a wide range of pH [17,18] and temperatures and organic solvents [19,20] and is not injurious to humans or animals. In addition, since the functionalization such as easily attaching metal ions to the DNA surface, there is a possibility that it can be applied to various fields of electronic devices [21,22]. Previous study reported plasmon-enhanced light absorption, photocurrent density and PCE by incorporating metallic NPs into mesoporous TiO_2_ (mp-TiO_2_) and electrolyte interface [23,24]. However, these plasmonic DSSCs have the critical issues when metal NPs were incorporated into the liquid electrolyte such as the aggregation of NPs [25]. For photoelectric applications, M13 bacteriophage-metallic nanostructure conjugates can be an excellent platform because M13 is easily fictionalized with metallic NPs by using phage display technology [26].

Herein, the M13 bacteriophage, which has improved the LSPR properties by anchoring of metallic silver (Ag) NPs, was added to the electrolyte of the solar cell. The electrolyte improved with an Ag NPs-embedded M13 bacteriophage (Ag@M13) showed a high absorbance in a long wavelength region of 500 nm or more, confirming that the plasmonic Ag NPs were uniformly oriented and aligned in nanoscale M13 bacteriophages. Furthermore, the FDSSCs were fabricated using an enhanced electrolyte supplemented with Ag@M13, which was demonstrated to improve the PCE by plasmonic enhancement effect. The optimized Ag@M13 enhanced FDSSC boasts effective electron extraction, unidirectional electron transportation, and suppressed charge recombination processes, resulting in a PCE of up to 5.80%, which was improved by 16.7% compared to that of the reference device with 4.97%.

## 2. Materials and Methods

### 2.1. Materials

Titanium wire (*Ф* 250 μm, 99.7%) was purchased from Sigma-Aldrich (Burlington, MA, U.S.). TiO_2_ paste (18NR-T) and iodine species (I^−^/I_3_^−^) based High Performance Electrolyte (HPE) were purchased from Greatcellsolar (Queanbeyan, Australia). Y123 dye (DN-F05Y) was purchased from Dyenamo (Stockholm, Sweden). Platinum wire (*Ф* 125 μm, 99.9%) was manufactured by a Wooillmetal (Yongin, South Korea).

### 2.2. Optical Simulation of Ag-BP-DSSC

The light scattering effect by Ag particles for a DSSC is simulated in the following way. The elemental samples of an electrolyte layer (60 μm) sealed with two glass substrates and an active layer (30 μm) that consists of a porous TiO_2_ (40% porosity) and dye mixture layer without the electrolyte were prepared on a glass substrate, and transmittance and reflectance of the samples were measured. Optical parameters (the effective refractive index, n-k values) of the DSSC active layer were extracted in consideration of TiO_2_ porosity, which was the result of the iterative process continuing the comparison of the measured transmittance and reflectance spectra with the calculated spectra several times [27]. Using a finite-difference time-domain method (FDTD) with the obtained optical parameters of the DSSC, the absorbance spectra in the active layer and Ag particles for different concentrations of Ag particles were calculated.

### 2.3. Fabrication of the FDSSC

The FDSSCs fabricated here are based on previous experiments [28,29]. A Ti wire was cleaned by deionized water, acetone, and isopropanol alcohol (IPA) sequentially for 10 min under the sonication. A compact-TiO_2_ (c-TiO_2_) layer was formed by electric heating with a current of 1.6 A for 10 s in air, allowing the oxidation of Ti wire and oxygen in the air to generate the c-TiO_2_ layer. The diluted TiO_2_ solution (1 g commercial TiO_2_ paste and 1 mL ethanol) was deposited by using the dip-coating technique with 10 mm s^−1^ for the withdraw rate and annealed at 120 °C for 3 m, and the cycle of dip-coating and annealing process was repeated five times to control the thickness of the mp-TiO_2_ layer. The PA layer consisting of the Ti/c-TiO_2_/mp-TiO_2_ layer was then crystallized by electric heating of 2 A for 10 m at ambient atmosphere. The PA was immersed in the Y123 dye solution dissolved in anhydrous ethanol (0.1 mM) for 4 h. The CE, Pt wire, was wound around the Y123 dye-sensitized PA so as not to damage the Y123 dye-sensitized PA and put into the Teflon tube. Then, after the electrolyte was injected, both ends of the tube were sealed.

### 2.4. Liquid Crystal (LC) Layering

Indium tin oxide (ITO) glass substrates with dimensions of 210 × 297 mm^2^ and a sheet resistance of 10 Ω sq^−1^ were ultrasonically cleaned using a semiconductor cleaning process. Polyimide (PI SE-7492; Nissan Chemical, Tokyo, Japan) was uniformly spin-coated onto the ITO-coated glass substrates to form liquid-crystal (LC) alignment layers. The PI layers were prebaked at 80 °C for 10 min and then imidized at 230 °C for 1 h. The thickness of the PI layer as an insulating film was set to approximately 100 nm. The PI layer was rubbed using a unidirectional rubbing method with a rubbing strength of 300 mm. Twisted-nematic (TN) LC cells were fabricated with a cell gap of 4.2 μm. To inject positive nematic LCs (MJ001929; ne = 1.5859, no = 1.4872, and Δε = 8.2, Merck, Darmstadt, German) effectively, a seal pattern was formed to create a vacuum inside the TN LC cell, and a vacuum injection method was used to inject liquid crystals using the capillary phenomenon and pressure difference at room temperature.

### 2.5. Characterization

Field-emission scanning electron microscopy (FE-SEM) images were obtained using a JEOL JSM-6700F field-emission scanning electron microscope (Tokyo, Japan). The optical transmittance was identified via UV-Vis-NIR spectroscopy (Cary 5000, Agilent Technologies, Santa Clara, CA, U.S.). The photocurrent density–voltage (*J*–*V*) characteristics of the FDSSCs were obtained by an electrometer (Keithley 2400, Keithley, Cleveland, OH, U.S.) under air-mass (AM) 1.5 illumination (100 mW cm^−^^2^) provided by a solar simulator (Oriel Sol3A Class AAA solar simulator, models 94043A, Newport, RI, U.S.). The calibrated light intensity was set to 1 sun using a standard silicon cell. The effective area of the device is defined as the project area transmitted by the mask, which is equal to the diameter of the photoanode multiplied by its length (1 cm). The external quantum efficiency (EQE) was measured using the incident photon-to-current conversion efficiency (IPCE) measurement system (QuantX 300, Oriel, Newport, RI, U.S.) with a 250 W quartz-tungsten-halogen lamp, an Oriel Cornerstone^TM^ 130 1/8 m monochromator operated in AC mode, an optical chopper, a lock-in amplifier, and a calibrated Si photodetector. The intensity of the sunlight for outdoor measurements was observed via a UV light meter (TM-208, Tenmars, Neihu, Taiwan). Electrochemical impedance spectroscopy (EIS) measurement was measured with an oscillation amplitude of 15 mV under dark conditions (Bio-Logic VMP-3, Seyssinet-Pariset, France) by using the open-circuit voltage and the frequency ranges from 1 Hz to 10 MHz. The experimental data were simulated using commercial Z-view software to estimate the values of each component of the corresponding equivalent circuits. The bending test was performed as a function of the number of bending cycles with a bending radius of 10 mm.

## 3. Results and Discussion

The metallic NPs, especially Ag NPs, are traditionally used to enhance the light-harvesting efficiency of optical devices due to their prominent LSPR properties. In addition, due to the unique charge selectivity of peptide receptors in M13 bacteriophage, the metallic NPs can be directly anchored onto the bacteriophage through charge-driven interactions without binder or surfactant [30]. Figure 1b shows high-resolution transmission electron microscopy (HR-TEM) (left) to investigate the morphology of Ag@M13 and an energy dispersive X-ray spectroscopy in scanning transmission electron microscopy (EDS-STEM) (right) to investigate its elemental mapping spectra. The nanoscale template of M13 bacteriophage nanostructures was well observed, which consists of nearly spherical NPs smaller than 20 nm in diameter. In addition, the elemental mapping spectra reveal the presence of the Ag NPs element onto an M13 bacteriophage nanostructure. From these results, it is confirmed that the Ag NPs are embedded by maintaining the nanoscale distance well in the bacteriophage template. Hence, the gap-plasmon effect can be expected as the metallic NPs maintain a constant nanoscale distance between them [31,32]. In addition, the gap-plasmon effect can be expected to have an additional effect of amplifying the absorption spectra in a long-wavelength region (about 600 nm) along with the existing effect of amplifying the absorption spectra in a short-wavelength region (about 300 nm) [33].

The normalized UV-Vis optical absorption spectra of the iodine based electrolytes as a function of the concentration of bare M13 bacteriophage and Ag@M13 bacteriophage are presented in Figure 1c,d, respectively. The electrolytes in which bare M13 bacteriophage (or Ag@M13) are added at concentrations of 5, 10, and 15 μL are referred to as M13-5, M13-10, and M13-15 (or Ag@M13-5, Ag@M13-10, and Ag@M13-15), respectively. Reference electrolyte (Ref) and the electrolyte with M13 and Ag@M13 had one absorption band at 370 nm. The absorption spectra of bare M13 bacteriophage increased from M13-5 to M13-10 and then decreased at M13-15. It was confirmed that the bare M13 bacteriophages as a function of the concentration added to the electrolyte hardly change each absorption spectra. In contrast, in the electrolyte to which Ag@M13 was added, it was confirmed that the absorption spectra improved in the long-wavelength region as the concentration of Ag@M13 increased and the full width at half maximum (FWHM) of the absorption band was also widened, as shown in the inset of Figure 1d. The LSPR properties, which are affected by the size of the metallic NPs, are limited to less than about 500 nm when the size of NPs becomes too small [30,34]. This research shows that the normalized absorbance has increased from 400 nm to the long-wavelength region, which means that the Ag NPs were aligned in a uniform arrangement on nanoscale template of M13 bacteriophage having uniform intervals in nanoscales. Therefore, the plasmonic Ag NPs were uniformly oriented and aligned in nanoscale bacteriophage, which contributed to overcoming the limitation of LSPR in smaller NPs.

The Ag particles included in the DSSC not only increase the absorbance of the active layer by scattering the incident light, which increases the photocurrent of the device, but also absorb the light the Ag particles absorb the light, which cannot contribute to the output photocurrent and lowers the overall photocurrent. Figure 2a shows the plasmonic effects that Ag particles absorb and scatter the light. The Ag particle effects increase with the size as shown in Figure 2b. Here, it is assumed that the size of Ag particles can be randomly distributed between 10–100 nm. As shown in Figure 2c, when the density of Ag particles increases, the amount of the light absorption by the Ag particles increases. In Figure 2d, when the density of Ag particles is ~1 × 10^9^/mm^3^, the absorption by the active layer can increase in the mid-range of visible light (500–700 nm), but above that concentration, the absorption decreases and even lowers compared to when the Ag is not present, due to the absorption by the Ag particles as shown in Figure 2c [35].

The photovoltaic performance of FDSSCs fabricated with or without improved by Ag@M13 was evaluated by investigating the current density–voltage (*J*–*V*) characteristic curves under AM 1.5 illumination, as shown in Figure 3a. In addition, their corresponding photovoltaic parameters are summarized in Table 1. Four parameters for identifying the characteristics of photovoltaic: open-circuit voltage (*V*_OC_), short-circuit current density (*J*_SC_), fill factor (FF), and PCE are important measures for analyzing the photovoltaic characteristics of FDSSCs. The conditions for FDSSCs in the form of (Ti wire/mp-TiO_2_/Y123 dye/Ag@M13 enhanced electrolyte/Pt wire) are shown in Appendix A. The *V*_OC_ increased from 0.65 to 0.66 V depending on the presence or absence of Ag@M13, but the change is very small. The *V*_OC_, which is the gap between the electrolyte and the Fermi level of the photoanode, showed insignificant changes even when Ag@M13 was added. This is interpreted as adding Ag@M13 into the electrolyte not affecting the energy level of the iodine based electrolyte. On one hand, the *J*_SC_ and the FF increased from the reference to Ag@M13-10 and then decreased at Ag@M13-15. Overall, the PCEs of FDSSCs enhanced by Ag@M13 enhanced electrolyte were higher than those of the bare electrolyte due to the increase in *J*_SC_ and FF. It is considered that the increase in *J*_SC_ means that Ag@M13 present in the electrolyte has a high probability of penetrating into the dye-sensitized photoanode with the mesoporous structure, which may have caused higher PCE. Furthermore, the increased FF was attributed to the increased shunt resistance (*R*_sh_), which represents the resistance to recombination, as well as the decreased series resistance (*R*_s_), which represents the overall resistance of the photoanode. The *R*_s_ is ideal for minimizing the decrease in current flow through the solar cell devices as its value is lower. In addition, the *R*_sh_ is ideal for preventing the recombination of photoexcited electrons into the electrolyte and maximizing the current flow to the external load as its values is higher. Thus, it is inferred that Ag@M13-10 has the lowest *R*_s_ value and the highest *R*_sh_ value, which would have affected the higher values of the *J*_SC_ and the FF. Hence, the Ag@M13-10 based FDSSC demonstrated the highest PCE of 5.80% with *J*_SC_ of 12.16 mA cm^−2^, *V*_OC_ of 0.66 V, and FF of 72.1%, whereas the bare electrolyte based FDSSC showed a PCE of 4.97% with *J*_SC_ of 10.71 mA cm^−2^, *V*_OC_ of 0.65 V, and FF of 71.7%.

The external quantum efficiency (EQE) spectra of the FDSSCs as a function of the concentration of Ag@M13 were investigated, and the results are shown in Figure 3b. In particular, the optimized Ag@M13-10 enhanced FDSSC showed a higher maximum EQE response than the other devices, which is consistent with the *J*_SC_ value in the *J*–*V* curve. The results shown *J*–*V* curves and EQDs above coherently demonstrate that there were more efficient charge extraction and less charge recombination in the Ag@M13 enhanced FDSSCs that the bare FDSSC. The improved performance by Ag@M13 enhanced electrolyte based FDSSCs benefited from the synergistic interaction between the plasmonic Ag NPs and M13 bacteriophages caused by the gap–plasmon effect amplified by the Ag NPs-embedded M13 bacteriophage nanostructures.

The dark *J*–*V* curves in Figure 3c show that the dark current curve slope varies as the voltage increases, which reflects different physical mechanisms. A, B, and C areas of the dark current have been reported to be associated with the shunt, recombination, and diffusion currents, respectively. In the A area, the dark current was mainly affected by the shunt current under a small applied bias voltage. It is shown that the *R*_sh_ value of Ag@M13-10 at the lowest slope in area A is very consistent with the *R*_sh_ value obtained from *J–V* curves in Figure 3a. As the bias voltage increases, the recombination current gradually increases similar to the diffusion current in the dark *J–V* characteristics, as shown in area B. It can be seen that the slope of area B increases more gently than that of area C. Moreover, the current increase in the area C is caused by the diffusion-dominated current. Indeed, in an area C higher than the built-in potential of DSSCs (about 0.7 V), the effect on the recombination current is negligible, and the curve is dominated only by the diffusion current limited by the *R*_s_ of the device [36]. Therefore, the Ag@M13 enhanced FDSSCs indicates that the leakage current decreased and the charge extraction and transport efficiency improved due to the relatively increased *R*_sh_ and decreased recombination current. Furthermore, the dark current density (*J*_0_) of the FDSSCs, which fabricated Ag@M13 as a function of concentration, was smaller than those of the bare electrolyte. This can be inferred as a phenomenon indicating that the dark current density was decreased due to the rapid transport of photoexcited electrons by the presence of Ag@M13 [37].

Power generation of wearable fiber electronic solar cells that depending on the daylight intensity should be considered for further development. Figure 3d shows the maximum power (*P*_MAX_) values corresponding to the time-dependent change of sunlight in an outdoor environment to evaluate the characteristics of the wearable applications. The *P*_MAX_ values of the Ag@M13 enhanced FDSSCs were verified by the following equation: *P*_MAX_ = *I* × *V* × FF. The sunlight intensity increased from the initial 400 W m^−2^ at 9:00 a.m. to 1400 W m^−2^ at 2:00 p.m., when the sun’s altitude reached its highest point. The light intensity gradually decreased to under 200 W m^−2^ at 5:00 p.m. As expected, the time-dependent change in daylight intensity and the corresponding *P*_MAX_ values of the solar cell devices showed the equivalent tendency.

EIS was performed under dark conditions at a bias voltage of 0.7 V in the frequency range of 0.1 Hz–1 MHz to accurately analyze the interfacial dynamics for the charge transfer behavior in Ag@M13 enhanced FDSSCs, as shown in the Nyquist plots in Figure 4a. The Nyquist plots simulated by the equivalent circuit diagram, as shown in Appendix A. The equivalent circuit, *R*_s_ + (CPE_ct1_//*R*_ct1_) + (CPE_ct2_//*R*_ct2_), consists of a series resistance (*R*_s_) and two charge-transfer resistances. The *R*_ct1_ and *R*_ct2_, which mean charge transfer resistance, are represented by the first smaller half-circle in the high-frequency region and the second larger half circle in the low-frequency region, respectively. These values are interpreted as interfacial resistance between CE/electrolyte and interfacial resistance between dye-sensitized PA/electrolyte, respectively. Moreover, the CPE_ct1_ and CPE_ct2_ represent the constant-phase elements for interfaces at CE/electrolyte and dye-sensitized PA/electrolyte, respectively [38]. The fitting values of *R*_s_, which is related to the transfer resistance of the photoanode, were 7.49, 6.85, 4.57, and 5.16 Ω for the Ref, Ag@M13-5, Ag@M13-10, and Ag@M13-15 enhanced FDSSCs, respectively, as shown in the inset of Figure 4a and Table 2. This tendency was very consistent with the *R*_s_ values obtained from the aforementioned *J*-*V* characteristic curves, and it can be seen that the *R*_s_ value is the lowest in Ag@M13-10 enhanced FDSSC. The *R*_ct1_ values showed very slight difference from 24.30 Ω at the Ag@M13-10 enhanced FDSSC, the lowest value, to 26.99 Ω at the Ref-FDSSC, the highest value. This tendency is also shown in CPE_ct1_. These results are interpreted that the Ag@M13 bacteriophages added to the electrolyte were not significantly correlated with CE. On the other hand, the *R*_ct2_ values representing the interfacial resistance between dye-sensitized PA/electrolyte were 129.0, 117.2, 53.1, and 75.8 Ω for the Ref, Ag@M13-5, Ag@M13-10, and Ag@M13-15 enhanced FDSSCs, respectively, showing clear difference. The lowest *R*_s_, *R*_ct1_, and *R*_ct2_ values observed for the Ag@M13-10 enhanced FDSSCs are most probably the consequence of the faster charge-transport process arising from the high probability that Ag@M13 penetrates into the dye-sensitized PA with the mesoporous structure, as mentioned above. Based on the improved charge transfer characteristics at the dye-sensitized PA/electrolyte interface, it can be confirmed that all the key parameters of the Ag@M13-10 enhanced FDSSCs could be enhanced, which resulted in high *J*_SC_ and PCE.

Catalytic activities and reaction kinetics of the Ag@M13 enhanced electrolyte were investigated by electrochemical characterization using cyclic voltammetry (CV) measurement, as shown in Figure 4b. The CV measurement of the enhanced electrolyte as a function of Ag@M13 was based on a three-electrode system consisting of two Pt wire electrodes as the working and counter electrode, respectively, and an Ag/AgCl as the reference electrode. The CV measurement was performed under the same conditions of applying the Pt wire as a working electrode and a counter electrode, and the analysis was focused on the catalytic properties of Ag@M13 injected into the electrolyte. The electrolyte used for CV was proceeded under the same condition as the electrolyte in the solar cell, and was used by adding 5, 10, and 15 μL of Ag@M13 to HPE. A pair of oxidation and reduction peaks can be identified, which corresponds to cathodic and anodic peaks indicated by reduction (Equation (1)) and oxidation (Equation (2)) reactions of iodide/triiodide, respectively [39]:(1)$$I_{3}^{-}+2e^{-}↔3I^{-}$$
(2)$$3I^{-}↔2e^{-}+I_{3}^{-}$$

Although the distribution and morphological shape of the peaks are not significantly different under all conditions, the intensity of the peak increased as the concentration of Ag@M13 increased, as shown in the inset of Figure 4b. This suggests that the interaction between Ag@M13 added to the electrolyte and CE is not large, but it can be considered that the injection of Ag@M13 into the electrolyte contributed to the catalytic activity to some extent. Thus, Ag@M13 added to the electrolyte exhibits higher current density on both the oxidation and reduction characteristics, indicating efficient charge transfer due to improved catalytic activities.

Weaving ability, durability, and water resistance are very important parameters in wearable device applications. The Ag@M13-10 enhanced FDSSCs exhibited excellent mechanical durability, as shown in Figure 4c. It can be seen that the normalized *η*/*η*_0_ of the Ag@M13-10 enhanced FDSSCs with a radius of about 10 mm showed a slight degradation in change to almost 80% even after 350 bending cycles. Even under extreme bending conditions such as radii 2.5, 5.0, and 7.5 mm, the normalized *η*/*η*_0_ was maintained without significant reduction, as shown in the inset of Figure 4c and Appendix A. Furthermore, the Ag@M13-10 enhanced FDSSCs exhibited excellent water resistance, as shown in Figure 4d. Repeated washing tests of Ag@M13-10 enhanced FDSSC were performed with an automatic cleaner, as shown in Appendix A. It was confirmed that the normalized *η*/*η*_0_ of the Ag@M13-10 enhanced FDSSC hardly decreased even after repeating the 16 iterations of the washing tests. These results indicate that Ag@M13 enhanced FDSSCs have excellent characteristics as wearable electronic devices. On the other hand, the Ag@M13 enhanced FDSSCs were connected in series and parallel to drive a liquid-crystal (LC) device for smart windows such as indoor lighting, as shown in Appendix A. The voltage and current density generated by the series and parallel connections of the Ag@M13 enhanced FDSSCs were about 2.3 V and 1.8 μA under indoor lighting of 172.8 W m^−2^, respectively, generating sufficient power for the operation of the LC device.

## 4. Conclusions

In conclusion, the Ag metallic NPs embedded M13 bacteriophages with the improved LSPR properties were supplemented to the electrolyte of the FDSSCs. It was confirmed that the Ag@M13 enhanced electrolyte showed high absorbance in a long wavelength region of 500 nm or more, and that the plasmonic Ag NPs were uniformly oriented and aligned in nanoscale M13 bacteriophages. The FDSSCs were fabricated using an Ag@M13 enhanced electrolyte, which was demonstrated to improve the PCE by plasmonic enhancement effect. The optimized Ag@M13 enhanced FDSSC boasts effective electron extraction, unidirectional electron transportation, and suppressed charge recombination processes, resulting in a PCE of up to 5.80%, which was improved by 16.7% compared to that of the reference device with 4.97%. In addition, the Ag@M13 enhanced FDSSCs maintained the PCE of >80% over 350 cycles of bending tests and >90% over 16 repetitions of washing tests.

## Figures and Tables

**Figure 1 nanomaterials-11-03421-f001:**
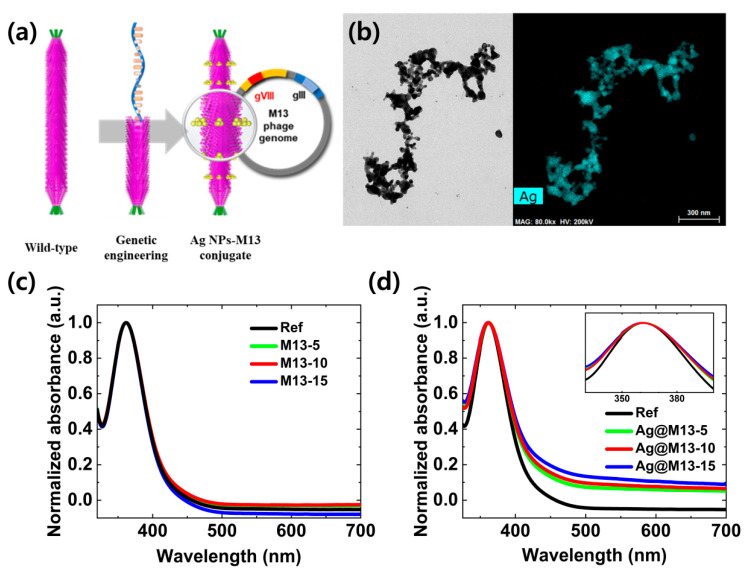
Morphological and optical characteristics of Ag@M13 bacteriophage: (**a**) schematic representation with structure on Ag@M13, (**b**) HR-TEM image (left) of morphology of Ag@M13 and EDS-STEM (right) to investigate its elemental mapping spectra, (**c**) normalized absorbance of as a function of M13 bacteriophage without Ag NPs, and (**d**) normalized absorbance of as a function of Ag@M13 bacteriophage. The inset is a high magnification of the main absorbance peak.

**Figure 2 nanomaterials-11-03421-f002:**
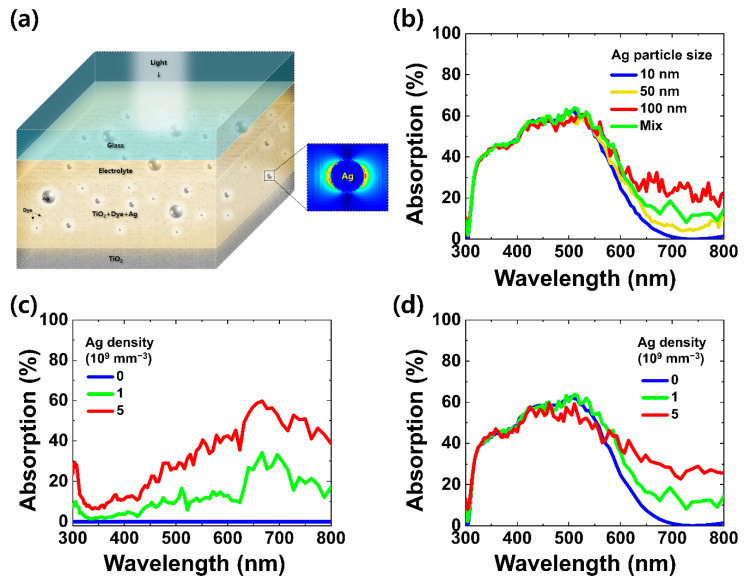
FDTD simulation of Ag-particle effects in the DSSC: (**a**) illustrated image of DSSC structure, (**b**) absorption spectra of active layer for the different Ag particle sizes, and absorption at (**c**) Ag particles and (**d**) at an active layer for the different Ag densities.

**Figure 3 nanomaterials-11-03421-f003:**
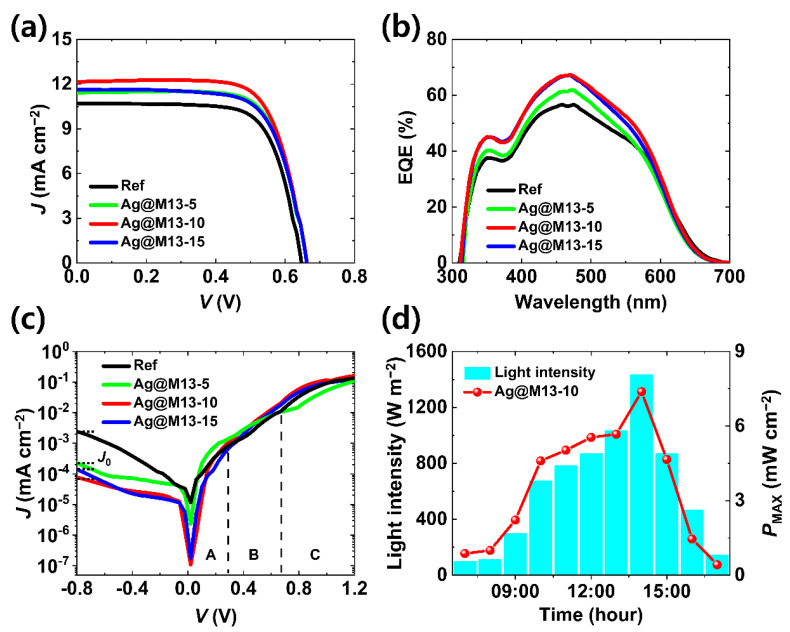
Photovoltaic properties of Ag@M13 enhanced FDSSCs: (**a**) comparison of *J–V* curves, (**b**) the IPCE spectra, (**c**) dark current profiles of FDSSCs as a function of Ag@M13, respectively, and (**d**) *P*_max_ values over time as an outdoor test.

**Figure 4 nanomaterials-11-03421-f004:**
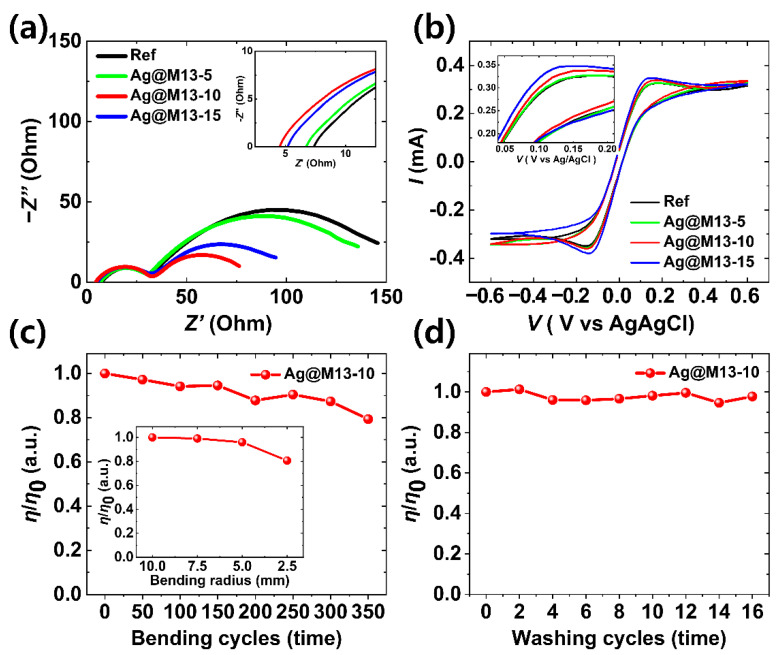
Electrochemical properties of the Ag@M13 enhanced FDSSCs: (**a**) Nyquist plots and onset point of Nyquist plots in the high-frequency region (inset), (**b**) CV characteristics. The three-electrode system, measured at a scan rate of 200 mV s^−1^, contains an Ag/AgCl as a reference electrode, Pt wires as working and counter electrodes, respectively. Normalized *η*/*η*_0_ of Ag@M13-10 enhanced FDSSCs as a function of (**c**) bending cycles and (**d**) washing cycles. The inset in Figure 4c is the normalized *η*/*η*_0_ under extreme bending conditions, such as radii 2.5, 5.0, and 7.5 mm.

**Table 1 nanomaterials-11-03421-t001:** Photovoltaic properties of FDSSCs as a function of Ag@M13.

	*V*_OC_ (V)	*J*_SC_ (mA cm^−2^)	FF (%)	PCE (%)	*R*_s_ (Ω cm^2^)	*R*_sh_ (Ω cm^2^)
Ref	0.65	10.71	71.7	4.97	7.13	6.97 × 10^4^
Ag@M13-5	0.66	11.44	71.8	5.47	5.36	8.90 × 10^4^
Ag@M13-10	0.66	12.16	72.1	5.80	4.32	1.26 × 10^5^
Ag@M13-15	0.66	11.61	69.9	5.39	5.29	9.43 × 10^4^

**Table 2 nanomaterials-11-03421-t002:** EIS parameters of SS-FDSSCs as a function of Ag@M13.

	*R*_s_ (Ω)	*R*_ct1_ (Ω)	CPE_ct1_ (F)	*R*_ct2_ (Ω)	CPE_ct2_ (F)
Ref	7.49	26.99	4.68 × 10^−5^	129.2	2.51 × 10^−3^
Ag@M13-5	6.85	26.24	4.66 × 10^−5^	117.2	2.75 × 10^−3^
Ag@M13-10	4.57	24.30	3.01 × 10^−5^	53.1	8.37 × 10^−3^
Ag@M13-15	5.16	25.01	4.27 × 10^−5^	75.8	5.74 × 10^−3^

## Data Availability

Not applicable.

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
