# Peer review of "Improved Light Harvesting of Fiber-Shaped Dye-Sensitized Solar Cells by Using a Bacteriophage Doping Method"

_nanomaterials, 2021, doi:10.3390/nano11123421_

Round 1
Reviewer 1 Report
Koo and co-authors have studied the performance of fiber-shaped dye-sensitized solar cells (FDSSCs) which can be enhanced by silver nanoparticles-embedded M13 bacteriophage (Ag@M13). The optimized FDSSC with Ag@M13 showed an enhanced PCE of up to 5.80%, which was improved by 16.7% compared to that of the control device with 4.97% without Ag@M13. The authors showed that Ag@M13 can improve the light absorption in the long wavelength domain of 500 nm or more due to the plasmonic Ag nanoparticles. The optimized Ag@M13 can improve effective electron extraction, unidirectional electron transportation, and suppressed charge recombination processes, thus improving the performance. The authors also showed that the FDSSCs with Ag@M13 have excellent durability and water resistance, which is promising for wearable electronics. In conclusion, this paper showed good results of FDSSCs and can be published in this journal in the present form.
Author Response
Koo and co-authors have studied the performance of fiber-shaped dye-sensitized solar cells (FDSSCs) which can be enhanced by silver nanoparticles-embedded M13 bacteriophage (Ag@M13). The optimized FDSSC with Ag@M13 showed an enhanced PCE of up to 5.80%, which was improved by 16.7% compared to that of the control device with 4.97% without Ag@M13. The authors showed that Ag@M13 can improve the light absorption in the long wavelength domain of 500 nm or more due to the plasmonic Ag nanoparticles. The optimized Ag@M13 can improve effective electron extraction, unidirectional electron transportation, and suppressed charge recombination processes, thus improving the performance. The authors also showed that the FDSSCs with Ag@M13 have excellent durability and water resistance, which is promising for wearable electronics. In conclusion, this paper showed good results of FDSSCs and can be published in this journal in the present form.
Response
We are very pleased that you have a good view of our research results.
We tried to make our research results look as good as possible and the results seem to have led to good reviews.
Finally, before this paper is published, we will once again carefully improve our English expression and style, and check that there are no typos. Once again we thank you for your good evaluation.

Reviewer 2 Report
This paper presents the implementation of Ag nanoparticles into organic dyes for fiber-shaped dye-sensitized solar cells. Despite including Ag nanoparticles improved slightly the solar cell efficiency, I could not see plausible reasons for such improvements, which is necessary to clarify this work.
For instance, in Fig 2, the Ag shows absorption contributions around 700 nm, however, the EQE in Fig 3 is increased around 450 nm ! Why is that ?
In Fig 4 lower resistivity is shown for Ag composites, Why this happens ?
I can see mostly an illustration about the obtained data without plausible mechanism for such improvements done by the Ag nanoparticles.
If the authors could provide a nice mechanism for the effect of Ag nanoparticles of the solar cell efficiency, I would be happy to see that and recommend this paper for publication.
Author Response
We have carefully read the reviewer’s comments describing our work and observations as original, timely, and important. We have revised our manuscript in accordance with the reviewer’s comments. These corrections are highlighted yellow in the revised manuscript. Please see our response in the attachment.

Round 2
Reviewer 2 Report
The authors have replied to my comments